# The SWGEDWGEIW from Soybean Peptides Reduces Insulin Resistance in 3T3-L1 Adipocytes by Activating p-Akt/GLUT4 Signaling Pathway

**DOI:** 10.3390/molecules28073001

**Published:** 2023-03-28

**Authors:** Guofu Yi, Xia Sang, Yuxia Zhu, Di Zhou, Shuibing Yang, Yue Huo, Yang Liu, Bushra Safdar, Xianyong Bu

**Affiliations:** 1Anhui Heat-Sensitive Materials Processing Engineering Technology Research Center, School of Biological Science and Food Engineering, Chuzhou University, Chuzhou 239000, China; 2Anhui Panpan Food Co., Ltd., Chuzhou 239064, China

**Keywords:** soybean peptide, SWGEDWGEIW, adipocyte, insulin resistance, lipid decomposition, p-Akt/GLUT4

## Abstract

Diabetes mellitus, a group of metabolic disorders characterized by persistent hyperglycemia, affects millions of people worldwide and is on the rise. Dietary proteins, from a wide range of food sources, are rich in bioactive peptides with anti-diabetic properties. Notably, the protective mechanism of the single peptide SWGEDWGEIW (TSP) from soybean peptides (SBPs) on insulin resistance of adipocytes in an inflammatory state was investigated by detecting the lipolysis and glucose absorption and utilization of adipocytes. The results showed that different concentrations of TSP (5, 10, 20 µg/mL) intervention can reduce 3T3-L1 adipocytes’ insulin resistance induced by inflammatory factors in a dose-dependent manner and increase glucose utilization by 34.2 ± 4.6%, 74.5 ± 5.2%, and 86.7 ± 6.1%, respectively. Thus, TSP can significantly alleviate the lipolysis of adipocytes caused by inflammatory factors. Further mechanism analysis found that inflammatory factors significantly reduced the phosphorylation (p-Akt) of Akt, two critical proteins of glucose metabolism in adipocytes, and the expression of GLUT4 protein downstream, resulting in impaired glucose utilization, while TSP intervention significantly increased the expression of these two proteins. After pretreatment of adipocytes with PI3K inhibitor (LY294002), TSP failed to reduce the inhibition of p-Akt and GLUT4 expression in adipocytes. Meanwhile, the corresponding significant decrease in glucose absorption and the increase in the fat decomposition of adipocytes indicated that TSP reduced 3T3-L1 adipocytes’ insulin resistance by specifically activating the p-Akt/GLUT4 signal pathway. Therefore, TSP has the potential to prevent obesity-induced adipose inflammation and insulin resistance.

## 1. Introduction

Since 1980, there has been an increase in the prevalence of overweight and obesity worldwide, and today, over one-third of the world’s population is categorized as overweight or obese [1]. Obesity is a prevalent metabolic disease characterized by excess accumulation of adipose tissue due to increased food intake and changes in lifestyle [2]. Obesity is the most common metabolic disorder in the world and a major risk factor for insulin resistance and diabetes [3]. Chronic systemic inflammation in obese patients arises from a localized immune response in visceral adipose tissue [4].

Low-grade chronic fat accumulation in adipose tissue (AT) inflammation (also known as meta-inflammation) is strongly associated with excess body fat mass and is characterized by infiltration and activation of pro-inflammatory macrophages and other immune cells that produce and secrete pro-inflammatory Cytokine [5]. The mild inflammation associated with obesity accelerates the infiltration of macrophages into adipose tissue, thereby accelerating the lipid decomposition of adipose tissue, leading to insulin resistance in the body [6]. Chronic systemic inflammation in obese patients arises from a localized immune response in visceral adipose tissue. Obesity-induced adipose inflammation is characterized by the hypertrophy of adipocytes and the accumulation and activation of adipose macrophages. The interaction between adipocytes and macrophages in the adipose tissue of obese people promotes the activation of macrophages into inflammatory macrophages (M1), thereby increasing the secretion of pro-inflammatory cytokines, such as interleukin-6 (IL-6), tumor necrosis factor α (TNF-α), and monocyte chemoattractant protein-1 (MCP-1), etc. [7]. Pro-inflammatory cytokines secreted by inflammatory macrophages lead to the degradation of insulin receptor substrate-1 (IRS-1), thereby impairing the insulin sensitivity of adipocytes [8]. Insulin receptor substrate-1/phosphoinositide 3-kinase/Akt (IRS-1/PI3K/Akt) is an essential molecular signaling pathway regulated by insulin; the degradation of IRS-1 may affect the IRS-1/PI3K/Akt pathway and expression of downstream-pathway glucose transporter 4 (GLUT4) [9]. As a key protein that regulates glucose uptake, the downregulation of GLUT4 eventually leads to the blockage of glucose uptake in adipocytes [10]. In addition, fatty tissue decomposes lipids in an inflammatory environment. It releases more free fatty acids, resulting in generally higher plasma-free fatty acid levels in obese people [11,12]. The increase in plasma-free fatty acid levels will further trigger the body’s insulin resistance (IR) [13].

Exacerbation of insulin resistance causes an increase in blood sugar level, which leads to diabetes, a common endocrine and metabolic disease [11]. The prevalence of diabetes is rapidly rising as a result of lifestyle changes and population aging [14]. It has become another chronic non-communicable disease that seriously endangers human health after cardiovascular and cerebrovascular diseases and tumors [15]. Its severe effects on the physical and mental health of patients, as well as its acute and chronic complications, especially those that involve multiple organs. These complications also result in high mortality and create disability [16]. In addition to lifestyle changes (such as dietary restriction, weight loss, and moderate exercise), diabetes interventions are widely used because of their ease of acceptance. At present, chemically synthesized drugs have the best effect, such as α-glucosidase inhibitors, biguanides, thiazolidinediones, insulin analogs, meglitinide, and phenylalanine derivatives, etc. However, the side effects of these drugs are relatively severe and commonly include liver and kidney damage, coma, diarrhea, hypoglycemia, lactic acidosis, etc. [17]. Additionally, resistance to these drugs can develop over time [18]. Therefore, finding novel, safe, and efficient treatments to prevent and treat diabetes is of utmost importance.

The World Health Organization has recommended to the globe a “modern comprehensive therapy (quintet)” for diabetes, with diet control ranking first [19]. Therefore, functional factors that regulate blood sugar have become research hotspots in the field of diabetes prevention and treatment. Dietary factors that affect blood glucose metabolism mainly include fatty acids, proteins and peptides, carbohydrates, antioxidant nutrients, minerals, and phytochemicals [20]. Studies have confirmed that islet cell dysfunction and decreased insulin sensitivity (insulin resistance) are the main causes of diabetes. Despite the availability of a wide range of medications for the treatment of diabetes, as well as the fact that diet and exercise regimens can help the disease stabilize and regress, these treatment strategies still fall short of completely and effectively inhibiting diabetes, hypertension, its pathogenesis, and the emergence of complications like blood vessels [21]. Therefore, it is of great significance to explore and develop new, safer, and more effective functional active substances from natural food resources. Proteins or peptides are the most ideal source because insulin itself is a polypeptide substance [22]. According to the law of similarity, it is better to find one or more peptides that synergize to inhibit and treat diabetes [23].

At present, the role of peptides in blood sugar regulation is mainly focused on endogenous active peptides, and plant foods like bitter melon, ganoderma, ginseng, and soybeans, as well as marine species, are the primary subjects of study on how natural diets regulate blood sugar [24]. Nowadays, soybeans are the top choice since they are extensively grown, easy to procure, rich in protein content, and have an optimal amino acid balance [25]. Soybean in particular has many potential health benefits in reducing chronic diseases such as obesity, cardiovascular disease, insulin resistance/type II diabetes, certain types of cancer, and immune disorders [26]. These physiological functions have been attributed to soy proteins, either intact soybean protein or, more commonly, functional or bioactive peptides derived from soybean processing. Due to these findings, a health claim that soy protein lowers the risk of coronary heart disease was approved in the United States, and a claim that soy protein lowers cholesterol levels was accepted in Canada [27]. Using different approaches, many soybean peptides (SBPs) have been identified to have various physiological functions, such as hypolipidemic [28], antihypertensive [29], and anticancer properties [30], as well as anti-inflammatory [31], antioxidant [32], and immune regulation [33]. Some soybean peptides, such as lunasin [34], have more than one of these properties and play a substantial role in the prevention of several chronic diseases. At the same time, studies have reported that SBPs can reduce insulin resistance in mice induced by a high-fat diet [35].

In previous studies, our team found that SBPs can inhibit lipopolysaccharide (LPS)-induced expression of inflammatory factors in RAW264.7 macrophages by inhibiting NF-κB and JNK/MAPK signaling pathways [36]. As well as SBPs protecting HepG2 cells from oxidative stress damage [37]. Another way SBPs inhibiting on apoptosis is via the activation of PI3K-AKT and inhibition of the apoptosis pathway [38]. SBPs regulate tryptophan hydroxylase (THP) and serotonin-N-acetyltransferase (AANAT) proteins to increase sleep [39]. SWGEDWGEIW (the single peptide, TSP) in SBPs reduces oxidative damage-mediated apoptosis in PC-12 cells by activating the SIRT3/FOXO3a signaling pathway [40]. TSP has a good inhibitory effect on hydrogen peroxide-induced inflammation, but it is unclear how TSP affects insulin resistance in adipocytes in this state of inflammation. 

However, the health effects of TSP on adipocytes are still unclear, and the mechanism by which TSP could be able to reduce the insulin resistance of adipocytes caused by obesity has not been revealed yet. This study intends to establish an in vitro model of simulating inflammation-induced insulin resistance in adipocytes to evaluate the effect of TSP on lipolysis and glucose utilization in adipocytes, with the aim of providing a theoretical basis for TSP to inhibit obesity and insulin resistance caused by obesity.

## 2. Results and Discussion

### 2.1. Effect of TSP on Insulin Resistance of 3T3-L1 Adipocytes

Figure 1 displays the experimental results on the effect of TSP on the glucose and lipid metabolism of adipocytes induced by inflammation. It can be seen from Figure 1A that TSP produced toxicity to adipocytes when the concentration was greater than or equal to 50 μg/mL but had no toxic effect on adipocytes when the concentration was 5–20 μg/mL. Therefore, the adipocyte insulin resistance model was further established to study the protective effect of 5–20 μg/mL TSP on adipocyte glucose absorption and utilization. It can be seen from Figure 1B that the 2-NBDG uptake level of 3T3-L1 adipocytes under inflammation induction was significantly reduced by 58.4% (*p* < 0.05) as compared to the blank group. At the same time, the sugar absorption level of 3T3-L1 adipocytes can be raised by TSP (5–20 μg/mL) in a dose-dependent manner. In addition, 3T3-L1 adipocytes treated with 5, 10, and 20 μg/mL TSP significantly increased the 2-NBDG uptake under inflammation induction, which was 34.2 ± 4.6%, 74.5 ± 5.2%, and 86.7 ± 6.1% higher than that of the inflammation induction group, respectively, indicating that TSP is protective against inflammation-induced disruption of glucose and lipid metabolism in adipocytes. Excessive lipid breakdown in adipocytes is generally regarded as the main cause of insulin resistance in the body [41]. Studies have shown that inflammation-induced lipolysis can induce a large amount of free fatty acid production, which further disrupts cellular insulin signaling and eventually leads to insulin resistance [42,43]. From Figure 1C,D, it can be seen that the lipid content of adipocytes induced by inflammation was reduced by 34.6 ± 3.9% compared to that of the blank group, while the release of glycerol was significantly increased by 83.5 ± 5.8%, indicating that inflammation induces lipid breakdown in adipocytes. However, TSP at 20 μg/mL significantly reversed the disorder of lipid content and glycerol release in adipocytes caused by inflammation (*p* < 0.05) and then inhibited lipid decomposition in adipocytes.

The Oil Red O staining results of adipocytes observed under a microscope are shown in Figure 2. It can be seen that after treatment with an inflammation-inducing medium, the shape of lipid droplets in 3T3-L1 adipocytes was significantly reduced, and the shape of lipid droplets in adipocytes returned to normal when the TSP concentration reached 20 μg/mL. The observed results of Oil Red O in 3T3-L1 adipocytes were consistent with the results of cell lipid content and glycerol release in Figure 1, indicating that TSP significantly reduced the lipolysis of 3T3-L1 adipocytes caused by inflammation. The research results show that TSP alleviates insulin resistance by improving the lipolysis of adipocytes caused by inflammation and then maintains the normal physiological function of adipocytes.

### 2.2. Effect of TSP on the Expression Level of Sugar Absorption-Related Proteins in 3T3-L1 Adipocytes

To further analyze the protective mechanism of TSP in improving insulin resistance in adipocytes caused by inflammation, Western Blot was used to quantitatively analyze the proteins in the pathways related to insulin regulation of sugar absorption. The experimental results are shown in Figure 3. It can be seen from Figure 3 that when the inflammatory medium induced insulin resistance in adipocytes, the expression levels of GLUT4 and p-Akt proteins in the cells were significantly reduced by 41.2 ± 4.3% and 22.1 ± 3.2% compared with those of the NC group (*p* < 0.05). GLUT4 and p-Akt are the main proteins in the insulin signaling pathway to regulate sugar absorption [44,45,46], in which p-Akt regulates the transfer of GLUT4 to the cell membrane and reduces glucose absorption and utilization [47]. Simultaneously, PI3K/Akt is a variety of active substances that reduce insulin resistance and important signaling pathways for increased sugar absorption [48]. It has been reported that a natural bioactive peptide isolated from soybean can significantly reduce insulin resistance in obese mice through the p-Akt pathway [49]. Similarly, this study found that the expression levels of GLUT4 and p-Akt proteins were significantly increased by 176.2 ± 12.6% and 182.3 ± 15.7% compared with those of the inflammation induction group (*p* < 0.05). The results of this study show that TSP may increase sugar absorption and reduce insulin resistance by activating the p-Akt/GLUT4 signaling pathway.

### 2.3. Analysis of TSP Improving Glucose and Lipid Metabolism in Adipocytes by Activating the p-Akt/GLUT4 Signaling Pathway

This work used LY as a PI3K inhibitor to examine if the protective effect of TSP on insulin resistance in adipocytes particularly stimulates the Akt signaling pathway, and it was verified by inhibiting PI3K activity to reduce the protein expression level of p-Akt. This can be used to verify whether TSP reduces insulin resistance and increases lipid breakdown through the PI3K-Akt signaling pathway. The experimental results are shown in Figure 4. From Figure 4A–C, it can be seen that after the pretreatment of adipocytes with PI3K inhibitor LY, the expressions of p-Akt and GLUT4 in the cells were significantly inhibited. However, TSP failed to increase the expression of p-Akt and GLUT4 in cells. Moreover, studies have shown that after p-Akt expression is inhibited, the efficacy of lipolysis in lowering insulin resistance is also diminished [44].

This study analyzed the lipid accumulation in adipocytes after LY intervention and explored whether the p-Akt/GLUT4 pathway activated by TSP was involved in the protective effect on lipid accumulation disorders in adipocytes. The effect of inhibitors on glucose uptake in adipocytes showed that TSP increased the uptake of 2-NBDG in 3T3-L1 adipocytes and was also significantly inhibited by LY (Figure 4D, *p* < 0.05). The results in Figure 5 show that lipid accumulation in adipocytes in the TSP 20 μg/mL group under LY interference was lower than that in the undisturbed group.

The results indicated that TSP was ineffective in reducing lipid accumulation in adipocytes due to restricted expression of the p-Akt/GLUT4 pathway. It was found that TSP can not only reduce sugar absorption by inhibiting lipolysis in adipocytes but also inhibit lipolysis by activating p-Akt/GLUT4-mediated sugar uptake.

A variety of phytochemicals in soybeans have been shown to increase sugar absorption and reduce insulin resistance by activating the PI3K/Akt/GLUT4 signaling pathway [50,51,52]. A peptide obtained from fermented soybean food has the potential for glucose utilization through the PI3K/Akt/GLUT4 pathway [53], and it has been demonstrated that TSP can exert biological activity by specifically activating PI3K/Akt. A study reported that a three amino acid soybean peptide phi-leu-val (FLV) can reduce the inflammatory response and insulin resistance in mature adipocytes. FLV inhibited the release of inflammatory cytokines (TNFα, MCP-1, and IL-6) from TNFα-stimulated adipocytes and co-cultured adipocytes/macrophages. This inhibition was mediated by the inactivation of the inflammatory signaling molecules c-Jun N-terminal kinase (JNK) and IjBa kinase (IKK) and the downregulation of IjBa in adipocytes [54].

The aforementioned studies relied on cells that have been verified to show that TSP lowers insulin resistance; however, more research is needed to determine whether TSP can enter the blood through the gastrointestinal tract when taken orally. Unlike the FLV reported in the research, there are only three amino acids, and such oligopeptides are generally less decomposed by pepsin. According to the composition of amino acids, 2–12 amino acids make up oligopeptides. Oligopeptides are not hydrolyzed once they reach the gastrointestinal tract and are easily absorbed. TSP is an oligopeptide with a molecular weight of about 1264. Tryptophan serves as the cleavage site for proteases in the gastrointestinal tract (especially pepsin), and TSP is rich in tryptophan. Hence, it is required to further investigate if TSP can pass through the gastrointestinal tract and enter the blood as well as whether it can meet the needs of cells when the concentration in the blood is 20 mg/mL.

## 3. Conclusions

This study found that TSP at a concentration of 20 μg/mL can effectively reduce insulin resistance in adipocytes caused by inflammation, and the absorption rate of 2-NBDG increases with the increase of TSP concentration, which proves that TSP has a certain effect on improving obesity-related insulin resistance and adipocyte lipolysis. The analysis results showed that TSP could significantly reduce the lipolysis of adipocytes caused by inflammation. The protein expression results showed that TSP increased the protein expression levels of p-Akt and GLUT4, but LY interfered with the reduction of TSP on adipocyte insulin resistance. TSP reduced inflammation-induced lipid accumulation disorders and insulin resistance in adipocytes by activating the p-Akt/GLUT4 signaling pathway. The findings of this study are expected to serve as a theoretical foundation for future investigations into the health benefits of peptides.

## 4. Materials and Methods

### 4.1. Materials

As shown in Figure 6, the single peptide of 99% purity was synthesized and bought from GL Biochem Ltd. (Shanghai, China), TSP; 3-isobutyl-1-methylxanthine (IBMX), dexamethasone, Insulin, LPS, and PI3K inhibitor (LY294002, abbreviated as LY) Purchased from Sigma Aldrich, St. Louis, MO, USA; Dulbecco’s Modified Eagle Medium (DMEM) high-glucose medium, calf serum (NBCS), fetal bovine serum (FBS), penicillin-streptomycin, and 0.25% trypsin were purchased from Gibco, USA; Glycerin Content Detection Kit was purchased from Nanjing Jiancheng Bioengineering Institute; Fluorescent Probe for Glucose Uptake and Transport (2-NBDG) Absorption Kit was purchased from American BioVision Company; anti-p-Akt (4060S), anti-Akt(9272S), anti-GLUT4 (2213), and anti-glyceraldehyde-3-phosphate dehydrogenase (GAPDH, AB-P-R001) were purchased from Cell Signaling Technology.

### 4.2. Preparation of Inflammation Induction Medium

RAW264.7 mouse macrophage cells were obtained from Peking Union Medical College and cultured using Dulbecco’s Modified Eagle Medium (DMEM) (containing 10% fetal bovine serum, 1% cyan-streptomycin diabody) at 37 °C. To prepare the cells for experiments, they were placed in a 5% (*v*/*v*) CO_2_ incubator and digested with 0.25% (*w*/*v*) trypsin solution. RAW264.7 cells were plated in a 96-well plate at a density of 6 × 10^4^ cells/mL, cultured in a group after 24 h. The control group and the LPS inflammation induction group were set up. The control group was replaced with fresh normal medium and continued to culture for 24 h. The inflammation model group was replaced with normal medium containing 1 μg/mL LPS, and cell inflammatory response was induced for 24 h [35]. The macrophage supernatant medium was collected and used as a blank medium and inflammation-inducing medium, respectively.

### 4.3. Induction, Differentiation, and Treatment of 3T3-L1 Preadipocytes

Peking Union Medical College provided the 3T3-L1 cells, which were cultured in DMEM and 10% volume fraction of NBCS medium at 37 °C and a 5% volume fraction of CO_2_. The 3T3-L1 cells were inoculated into culture plates at a confluence of 60%, cultured with growth medium (DMEM+10%NBCS) until the cells were completely confluent, and then cultured for an additional 2 days to allow the cells to become fully confluent. Production of contact inhibition was as follows: induction solution I was added (high glucose DMEM + 10% FBS, 10 μg/mL insulin, 1 μmol/L dexamethasone, 0.5 mmol/L IIBMX medium) and cultured for 2 days, then replaced with induction solution II (high glucose DMEM + 10% FBS, 10 μg/mL insulin medium), which cultured for an additional 2 days. Finally, it was shifted to a high-glucose medium (DMEM + 10%FBS) for 2 days to obtain differentiated mature adipocytes. Differentiated mature adipocytes were treated with 5, 10, and 20 μg/mL TSP for 24 h; the inflammation-inducing medium was replaced after 24 h; and then various indicators were detected [55].

### 4.4. Cell Viability Detection

The 3T3-L1 adipocytes were cultured using Dulbecco’s Modified Eagle Medium (DMEM) complete medium (containing 10% fetal bovine serum (FBS), 1% cyan-streptomycin diabody) at 37 °C. The cells were cultured in a 5% (*v*/*v*) CO_2_ incubator and digested with 0.25% (*w*/*v*) trypsin solution, which was plated in 96-well plates at a density of 6 × 10^4^ cells/mL. First, 10 mL of fresh DMEM containing different concentrations of TSP (0, 5, 10, 20, 50, 100, and 200 μg/mL) was cultured for 24 h. Cell viability assays were performed according to the kit instructions using the CCK-8 method [56]. The absorption of each well in the plate was recorded at 450 nm using an Infinite 200 Pro NanoQuant plate reader to evaluate the survival rates of the 3T3-L1 cells.

### 4.5. PI3K Inhibitor Interference Treatment

When the 3T3-L1 adipocytes were differentiated and matured, TSP was added to treat for 24 h and then shifted to a fresh medium containing 1 μmol/L LY for 24 h. After the supernatant was absorbed, the PBS was washed twice, and the inflammation-inducing medium was changed for 24 h. Lipid decomposition and glucose absorption indicators were detected [57].

### 4.6. Detection of Glucose Uptake and Lipid Decomposition in Adipocytes

The cellular glucose uptake level was evaluated by detecting the fluorescence intensity of 2-NBDG. After 3T3-L1 mature adipocytes were subjected to different treatments, the cell supernatant culture fluid was collected, and the degree of lipid decomposition was tested by detecting the glycerol content. According to the operating instructions of the 2-NBDG absorption kit, the glucose uptake enhancer was added to the DMEM medium at a volume ratio of 1:100, and the cells continued to replace the DMEM (containing glucose uptake enhancer) medium containing 100 μg/mL 2-NBDG, cultured for 30 min, remove the supernatant and wash it twice with PBS. Cells were then cultured in a 4% mass fraction of paraformaldehyde solution for 5 h, then stained with DAPI solution for 5 min. The glucose uptake capacity of adipocytes was expressed by the ratio of the fluorescence intensity of adipocytes at the excitation/emission wavelengths of 465/540 nm (2-NBDG) and 358/461 nm (DAPI) and normalized to the inflammation-inducing group [58].

### 4.7. Oil Red O Staining

After treating 3T3-L1 mature adipocytes with the methods described above, remove the supernatant and wash it with PBS three times. Fix the cells with 4% paraformaldehyde solution by volume for 1 h, remove the formaldehyde solution, and rinse them twice with PBS. Mordant was treated with 60% isopropanol for 15 s and stained with Oil Red O reagent for 30 min. After the staining was terminated, the dye solution and the excess Oil Red O reagent were removed by washing with 60% isopropanol for 15 s. After washing thrice with PBS, samples were observed under the microscope, and pictures with a 5 μm size were taken. A total of 1000 μL of isopropanol was added to each well, and the well was shaken for 5 min to wash out Oil Red O. A total of 200 μL of the isopropanol solution was used to wash out the Oil Red O from each well, transferred to a 96-well plate, and the absorbance measured at 490 nm [59].

### 4.8. Western Blot Experiment

The 3T3-L1 adipocytes were lysed with ice-cold RIPA Lysis (Thermo Fisher Scientific, Shanghai, China) and incubated on ice for 30 min to extract proteins. The supernatant was collected by centrifugation at 14,000× *g* for 15 min at 4 °C for further testing. The protein concentration of each sample was measured by the BCA Protein Assay Kit (Thermo Fisher Scientific, Shanghai, China). Then the supernatant was boiled in 5× loading buffer for 10 min and electrophoresed on a 10% (*w*/*v*) sodium dodecyl sulfate-polyacrylamide gel (SDS-PAGE). The protein bands were transferred to an NC membrane (Millipore, Burlington, MA, USA) and blocked with 5% (*w*/*v*) skim milk for 1 h at room temperature. After three washes in Tris-buffered saline containing Tween 20 (TBST), the membranes were incubated with designated primary antibodies (p-Akt, Akt, GLUT4) at 4 °C overnight then washed again with TBST 5 times, and the secondary antibody conjugated with peroxidase was incubated at room temperature with 1:5000 dilution. After washing the membrane, protein was detected using ECL (Millipore, Burlington, NJ, USA) with GAPDH as a loading reference [37].

### 4.9. Data Analysis

The experiments were repeated three times, and data were processed by GraphPad Prism 8.0 and SPSS 16.0. The experimental results were expressed as mean ± standard deviation, and the comparison between groups was conducted by one-way ANOVA and Duncan. Different letters indicated that different samples had significant differences on the same index (*p* < 0.05).

## Figures and Tables

**Figure 1 molecules-28-03001-f001:**
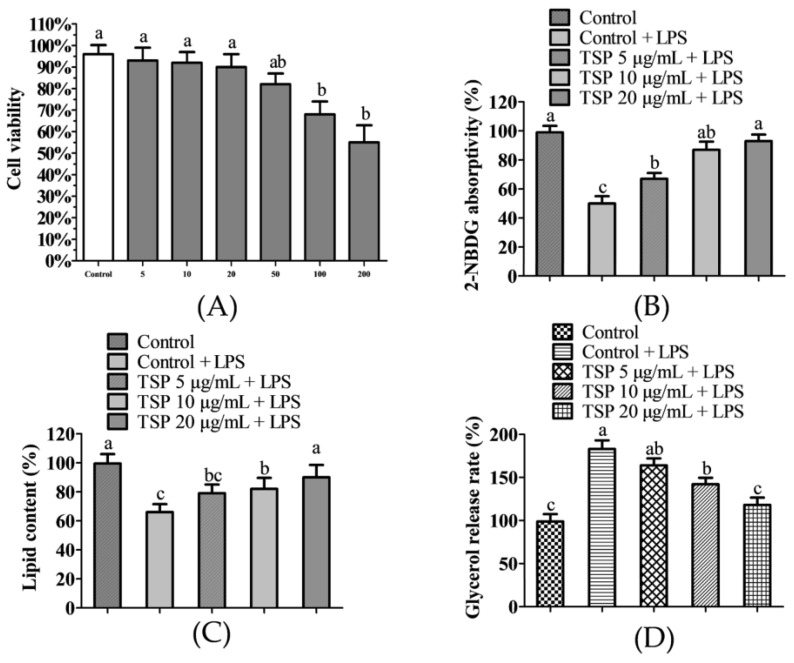
Effects of TSP on the viability of 3T3-L1 cells (**A**), the effect of TSP on glucose uptake levels in 3T3-L1 adipocytes was evaluated using 2-NBDG absorption values (**B**), the effect of TSP on the lipid level of 3T3-L1 adipocytes under inflammatory conditions was evaluated by absorption values (**C**), the effect of TSP on glycerol release from 3T3-L1 adipocytes under inflammatory conditions was evaluated by absorption values (**D**). Data are presented as mean ± SD (n = 3); Results marked with the same letters were not significantly different (*p* < 0.05).

**Figure 2 molecules-28-03001-f002:**
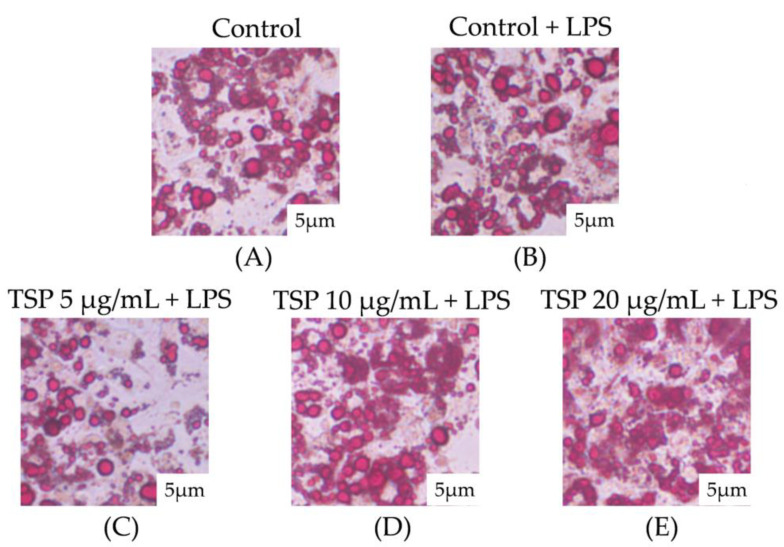
Oil red O staining of the effect of TSP on inflammatory-induced lipid accumulation in adipocytes: (**A**) is a blank control group; (**B**) is a positive control group under inflammatory conditions; (**C**) is TSP concentration 5 μg/mL low dose group; (**D**) is TSP concentration 10 μg/mL Medium dose group; (**E**) is TSP concentration 20 μg/mL high dose group.

**Figure 3 molecules-28-03001-f003:**
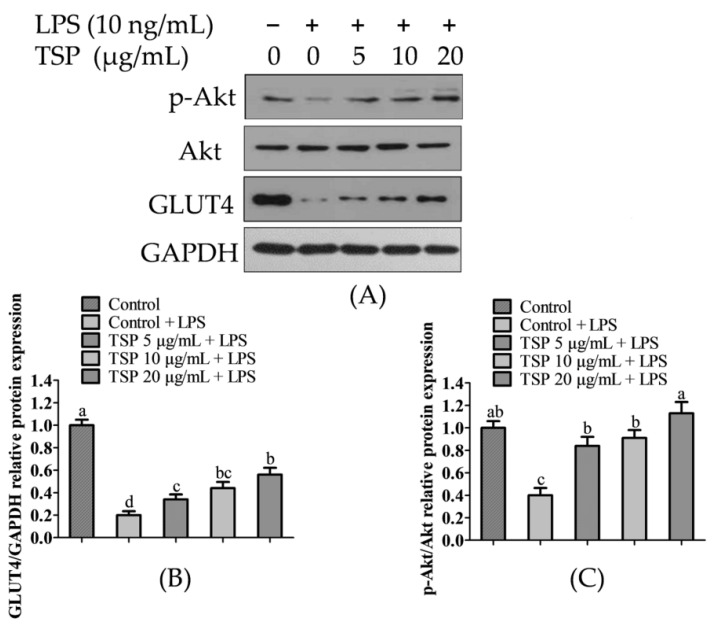
TSP can alleviate the inflammatory-induced insulin resistance of adipocytes. (**A**) The expression of p-Akt, Akt, and GLUT4 proteins was tested by western blotting. (**B**) GLUT4/GAPDH relative protein expression. (**C**) p-Akt/Akt relative protein expression. Data are presented as mean ± SD (n = 3); Results marked with the same letters were not significantly different (*p* < 0.05).

**Figure 4 molecules-28-03001-f004:**
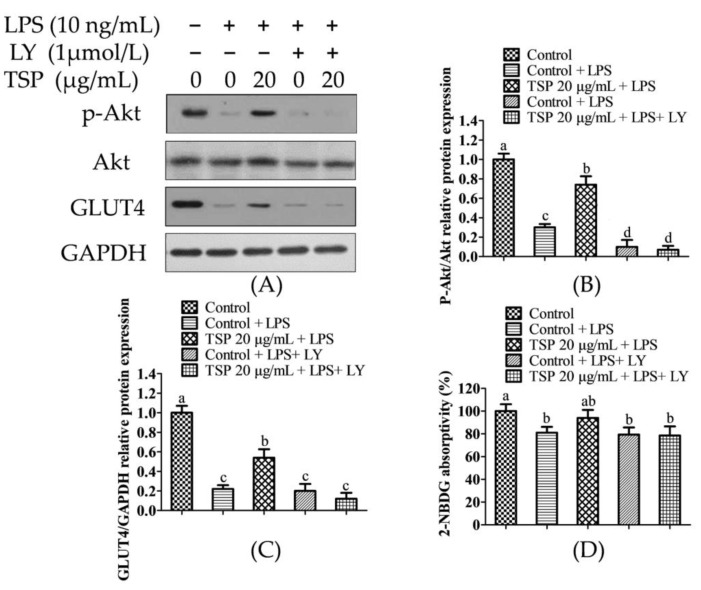
TSP regulates the effect of inflammation-induced insulin resistance in adipocytes under LY conditions. (**A**) The expression of p-Akt, Akt, and GLUT4 proteins was tested by western blotting. (**B**) GLUT4/GAPDH relative protein expression. (**C**) p-Akt/Akt relative protein expression. (**D**) 2-NBDG absorptivity. Data are presented as mean ± SD (n = 3); Results marked with the same letters were not significantly different (*p* < 0.05).

**Figure 5 molecules-28-03001-f005:**
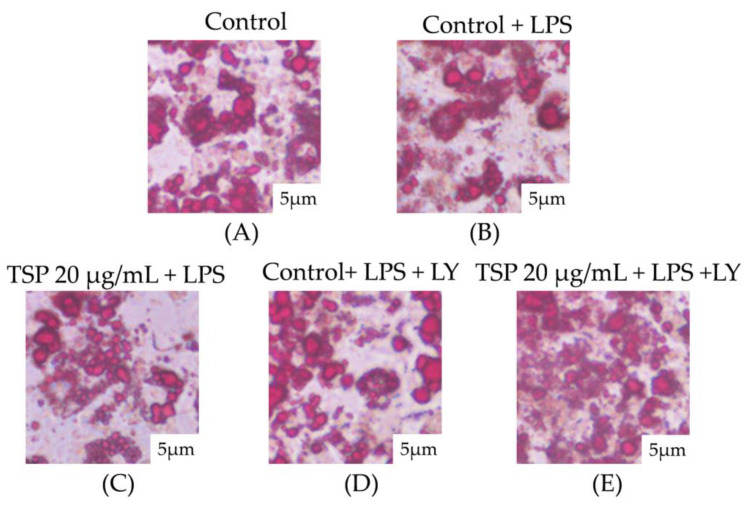
Oil red O staining diagram of the effect of TSP on lipid accumulation in adipocytes under inflammatory stress and inhibitor LY: (**A**) blank control group, (**B**) positive control group under inflammatory conditions, (**C**) Effect of TSP (20 μg/mL) on positive control group, (**D**) Positive control group with inhibitor LY; (**E**) Effect of TSP on positive control group supplemented with inhibitor LY.

**Figure 6 molecules-28-03001-f006:**
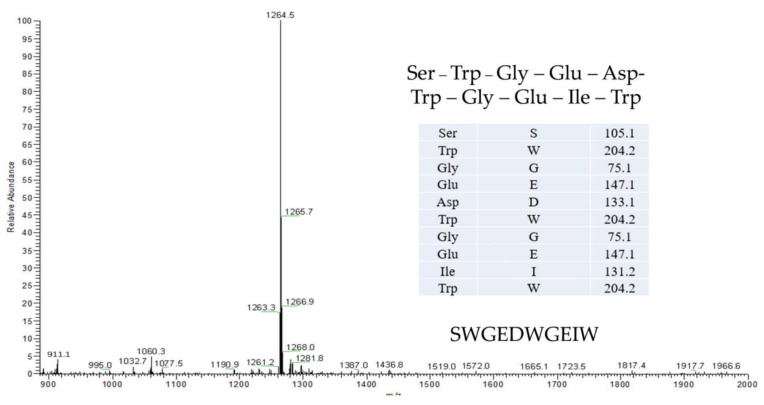
Mass spectra of TSP.

## Data Availability

The data presented in this study are available on request from the corresponding authors.

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
