# Peer review of "The SWGEDWGEIW from Soybean Peptides Reduces Insulin Resistance in 3T3-L1 Adipocytes by Activating p-Akt/GLUT4 Signaling Pathway"

_molecules, 2023, doi:10.3390/molecules28073001_

Round 1
Reviewer 1 Report
In this manuscript (MS; molecules-2238654) entitled “The SWGEDWGEIW from soybean peptides enhances insulin resistance in 3T3-L1 adipocytes by activating p-Akt/GLUT4 signaling pathway”, the authors propose that soybean peptide (SP) ameliorates inflammatory stress. The key finding is that SP increases Akt/GLUT4 signaling. Soy peptides have previously been studied to reduce inflammatory responses and improve insulin-stimulated glucose uptake (PMID; 27322965), with similar results in this MS. In addition, the data are not novel as there is no comparison to existing results. Furthermore, the title of the manuscript is “Soybean Peptides Enhances Insulin Resistance”, which indicates that it should not be accepted due to the discrepancy between the title and the results. I will comment some points to the authors below.
1. The title does not match the content of the results.
2. There are many spelling mistakes in the MS. For example, 3T3-L1 and 3t3-L1 are mixed up in the MS. The authors should have the English text proofread.
3. What does TSP stand for? The reviewer can see that SP is soybean peptides, but the reviewer doesn't know what “T” is.
3. The authors add 20 mg/mL TSP to the cells, but is 20 mg/mL an adequate blood concentration of SP? To what extent is SP absorbed from the intestinal tract? This information is very important when dealing with food factors, and the results of the authors’ data are not physiological if this information is not available. The authors should discuss blood levels of TSP in the Results and Discussion sections.
4. ANOVA is a method of analysis of variance, and a multiple comparison test is required to analyze significant differences between different alphabets. The authors should also describe the multiple comparisons method in detail in the M&M sections.
5. Fig. 2c and Fig. 3 duplicate the same data. In addition, the authors describe that the shape of lipid droplets in 3T3-L1 adipocytes was significantly reduced. The authors should measure the size of each lipid droplet to demonstrate the miniaturization of lipid droplets.
6. The font size of the figure is too small. The authors should either increase the font size or provide higher resolution images.
7. Insulin promotes the membrane translocation of GLUT4 by activating Akt signaling. Therefore, the authors should to detect cell membrane-localized GLUT4 by Western blotting.
Author Response
Response to Reviewer 1 Comments
Dear honoured reviewers,
We really appreciate you and the reviewers for your comments on our Manuscript: ““The SWGEDWGEIW from soybean peptides enhances insulin resistance in 3T3-L1 adipocytes by activating p-Akt/GLUT4 signaling pathway” We express our sincere gratitude and thankfulness for your time and precision in reviewing our manuscript. The responses to the comments are as follows. For your kind information, we have carefully dealt with the comments of the reviewers as follows: All the places which we changed have already been marked yellow in our paper, for the purpose of highlight. We hope the revised manuscript meets the standard of publication. Thank you!
In this manuscript (MS; molecules-2238654) entitled “The SWGEDWGEIW from soybean peptides enhances insulin resistance in 3T3-L1 adipocytes by activating p-Akt/GLUT4 signaling pathway”, the authors propose that soybean peptide (SP) ameliorates inflammatory stress. The key finding is that SP increases Akt/GLUT4 signaling. Soy peptides have previously been studied to reduce inflammatory responses and improve insulin-stimulated glucose uptake (PMID; 27322965), with similar results in this MS. In addition, the data are not novel as there is no comparison to existing results. Furthermore, the title of the manuscript is “Soybean Peptides Enhances Insulin Resistance”, which indicates that it should not be accepted due to the discrepancy between the title and the results. I will comment some points to the authors below.
Point 1: The title does not match the content of the results.
Response 1: Thank you very much for the comments and suggestions. We have changed the title of the manuscript to Reducing Insulin Resistance. Please check.
Point 2: There are many spelling mistakes in the MS. For example, 3T3-L1 and 3t3-L1 are mixed up in the MS. The authors should have the English text proofread.
Response 2: We have corrected spelling errors in the whole manuscript as per suggestion Please check.
Point 3: What does TSP stand for? The reviewer can see that SP is soybean peptides, but the reviewer doesn't know what “T” is.
Response 3: Thank you very much for the comments and suggestions. This T represents the first letter of The, and TSP is the acronym for the single peptide. We know this is non-standard, but SWGEDWGEIW is too long to write, so TSP is used instead. For clarification, it has been mentioned in the manuscript (Line).
Point 4: The authors add 20 mg/mL TSP to the cells, but is 20 mg/mL an adequate blood concentration of SP? To what extent is SP absorbed from the intestinal tract? This information is very important when dealing with food factors, and the results of the authors’ data are not physiological if this information is not available. The authors should discuss blood levels of TSP in the Results and Discussion sections.
Response 4: Thanks for your suggestion. We are currently studying the concentration added to the cells. At present, the concentration added to the cells is 20 mg/mL. In terms of blood concentration and intestinal absorption, the research needs to be carried out in small animal experiments. This part of the research will be carried out in future research exploration. We have discussed the study of TSP blood levels in the Results and Discussion section, please see lines 228-247.
Point 5: ANOVA is a method of analysis of variance, and a multiple comparison test is required to analyze significant differences between different alphabets. The authors should also describe the multiple comparisons method in detail in the M&M sections.
Response 5: Thank you for your advice. We have made additions and modifications to the analysis method, please see lines 366-370.
Point 6: Fig. 2c and Fig. 3 duplicate the same data. In addition, the authors describe that the shape of lipid droplets in 3T3-L1 adipocytes was significantly reduced. The authors should measure the size of each lipid droplet to demonstrate the miniaturization of lipid droplets.
Response 6: Thank you very much for the comments and suggestions. We just compared the previous cells as long as there was a decline to calculate.
Point 7: The font size of the figure is too small. The authors should either increase the font size or provide higher resolution images.
Response 7: Thank you very much for the comments. We have enlarged the font size in the figure. Please check.
Point 8: Insulin promotes the membrane translocation of GLUT4 by activating Akt signaling. Therefore, the authors should to detect cell membrane-localized GLUT4 by Western blotting
Response 8: Thank you very much for the comments. The intracellular transporter we write about in this article refers to the transporter on the cell membrane. Since this protein only exists on the cell membrane and not within the cell. We have corrected this wording and the sentence has been modified, this is our mistake. Thank you again for your valuable comments.

Reviewer 2 Report
The manuscript describes the beneficial effects of a short peptide from soybean (TSP) on the insulin resistance in adipocytes 3T3-L1 conditioned with inflammation-inducing-medium. The authors demonstrated that two hallmarks of insulin resistance, reduced glucose uptake and increased lipolysis, could be ameliorated by TSP in a dose-dependent manner. The authors provided further evidence showing that the effects of TSP were related with the PI3L/Akt signal pathway. The manuscripts merits publication, however, there are many places need to be improved.
(1) My first concern is the Oil Red O staining results. In Figure 3 (a) and (b), the authors claimed that the lipid droplets were significant reduced after treatment with inflammation-inducing-medium, however, the images were not obviously to me. Similarly in Figure 6 (a) and (b), without quantitative analysis of the images, the pictures couldn’t demonstrate the effects on lipid accumulation the authors have claimed.
(2) No reference to the method to assay the glycerol content in medium.
(3) Line 204 ‘significantly reduced by 98.4%’? From the figure, it is reduced by roughly half.
There are many English errors in the writing. For some text, it is not comprehensible, where I can only figure out its meaning by looking at the corresponding figures (e.g. line 217-219, line 268, line 270 etc.)
Author Response
Response to Reviewer 2 Comments
Dear honoured reviewers,
We really appreciate you and the reviewers for your comments on our Manuscript: ““The SWGEDWGEIW from soybean peptides enhances insulin resistance in 3T3-L1 adipocytes by activating p-Akt/GLUT4 signaling pathway” We express our sincere gratitude and thankfulness for your time and precision in reviewing our manuscript. The responses to the comments are as follows. For your kind information, we have carefully dealt with the comments of the reviewers as follows: All the places which we changed have already been marked yellow in our paper, for the purpose of highlight. We hope the revised manuscript meets the standard of publication. Thank you!
The manuscript describes the beneficial effects of a short peptide from soybean (TSP) on the insulin resistance in adipocytes 3T3-L1 conditioned with inflammation-inducing-medium. The authors demonstrated that two hallmarks of insulin resistance, reduced glucose uptake and increased lipolysis, could be ameliorated by TSP in a dose-dependent manner. The authors provided further evidence showing that the effects of TSP were related with the PI3L/Akt signal pathway. The manuscripts merits publication, however, there are many places need to be improved.
Point 1: My first concern is the Oil Red O staining results. In Figure 3 (a) and (b), the authors claimed that the lipid droplets were significant reduced after treatment with inflammation-inducing-medium, however, the images were not obviously to me. Similarly in Figure 6 (a) and (b), without quantitative analysis of the images, the pictures couldn’t demonstrate the effects on lipid accumulation the authors have claimed.
Response 1: Thank you very much for the comments and suggestions. We just compared the control group cells as long as there is a decline and can be calculated. There is further control through the determination of lipid content. The measurement of lipid content provides further control. Determining the lipid content allows for further control.
Point 2: No reference to the method to assay the glycerol content in medium.
Response 2: Thank you very much for the comments and suggestions. The Method 4.6 describes the procedure for determining the glycerol content in the medium in line 326-337.
Point 3: Line 204 ‘significantly reduced by 98.4%’? From the figure, it is reduced by roughly half.
There are many English errors in the writing. For some text, it is not comprehensible, where I can only figure out its meaning by looking at the corresponding figures (e.g. line 217-219, line 268, line 270 etc.).
Response 3: Thank you very much for the comments and suggestions. We have made changes based on your suggestions. Please see lines158-160, 207-213.

Reviewer 3 Report
Manuscript Title: The SWGEDWGEIW from Soybean Peptides enhances insulin resistance in 3t3-L1 adipocytes by activating p-Akt/GLUT4 signaling pathway
Manuscript Number: molecules-2238654
Article Type: Article
Comments:
The manuscript “The SWGEDWGEIW from Soybean Peptides enhances insulin resistance in 3t3-L1 adipocytes by activating p-Akt/GLUT4 signaling pathway” by Guofu Yi and co-workers, developed small peptide SWGEDWGEIW from Soybean which enhances insulin resistance by activating p-Akt/GLUT4 signaling pathway.
Overall, I thoroughly enjoyed reading this draft along with the supporting information. This work was well composed with a good balance of supportive experimental data. Good to see the data for the peptide and its activities. The mechanism well studied for the single peptide SWGEDWGEIW from Soybean peptide on insulin resistance of adipocytes in an inflammatory state with supporting data. The results described in this draft are more appropriate and convincing with the experimental data. I strongly accept this draft for publication without any revision. Furthermore, this work is high relevance and suitable for publication in the ‘Molecules’ journal.
Author Response
Response to Reviewer 3 Comments
Dear honoured reviewers,
We really appreciate you and the reviewers for your comments on our Manuscript: ““The SWGEDWGEIW from soybean peptides enhances insulin resistance in 3T3-L1 adipocytes by activating p-Akt/GLUT4 signaling pathway” We express our sincere gratitude and thankfulness for your time and precision in reviewing our manuscript. The responses to the comments are as follows. For your kind information, we have carefully dealt with the comments of the reviewers as follows: All the places which we changed have already been marked yellow in our paper, for the purpose of highlight. We hope the revised manuscript meets the standard of publication. Thank you!
The manuscript “The SWGEDWGEIW from Soybean Peptides enhances insulin resistance in 3t3-L1 adipocytes by activating p-Akt/GLUT4 signaling pathway” by Guofu Yi and co-workers, developed small peptide SWGEDWGEIW from Soybean which enhances insulin resistance by activating p-Akt/GLUT4 signaling pathway.
Overall, I thoroughly enjoyed reading this draft along with the supporting information. This work was well composed with a good balance of supportive experimental data. Good to see the data for the peptide and its activities. The mechanism well studied for the single peptide SWGEDWGEIW from Soybean peptide on insulin resistance of adipocytes in an inflammatory state with supporting data. The results described in this draft are more appropriate and convincing with the experimental data. I strongly accept this draft for publication without any revision. Furthermore, this work is high relevance and suitable for publication in the ‘Molecules’ journal.
Response: Thank you very much for the comments and suggestions.

Reviewer 4 Report
There are numerous grammatical, language and typographical errors in the manuscritp which needs correction.
Refrences should be cited in methodology section for each protocol utilized.
The abstract should be rewritten as it seems a compilation of results and no background of the study and signifcance have been mentioned.
Scale should be provided for all oil red O staining images.
Author Response
Response to Reviewer 4 Comments
Dear honoured reviewers,
We really appreciate you and the reviewers for your comments on our Manuscript: ““The SWGEDWGEIW from soybean peptides enhances insulin resistance in 3T3-L1 adipocytes by activating p-Akt/GLUT4 signaling pathway” We express our sincere gratitude and thankfulness for your time and precision in reviewing our manuscript. The responses to the comments are as follows. For your kind information, we have carefully dealt with the comments of the reviewers as follows: All the places which we changed have already been marked yellow in our paper, for the purpose of highlight. We hope the revised manuscript meets the standard of publication. Thank you!
Point 1: There are numerous grammatical, language and typographical errors in the manuscritp which needs correction.
Response 1: Thank you very much for the comments and suggestions. We have corrected grammatical, linguistic, and typographical errors in the whole manuscript. Please check.
Point 2: Refrences should be cited in methodology section for each protocol utilized.
Response 2: Thank you very much for the comments and suggestions. We have added corresponding references for each experimental protocol. Please check the methodology section.
Point 3: The abstract should be rewritten as it seems a compilation of results and no background of the study and significance have been mentioned.
Response 3: Thank you very much for the comments and suggestions. We have rewritten the abstract according to your suggestion.
Point 4: Scale should be provided for all oil red O staining images.
Response 4: Thanks for your suggestion. We have provided scale bars for all Oil Red O stained images.

Round 2
Reviewer 4 Report
The article may be published after thorough english editing.
Author Response
Response to Reviewer 4 Comments
Dear honoured reviewers,
We really appreciate you and the reviewers for your comments on our Manuscript: ““The SWGEDWGEIW from soybean peptides enhances insulin resistance in 3T3-L1 adipocytes by activating p-Akt/GLUT4 signaling pathway” We express our sincere gratitude and thankfulness for your time and precision in reviewing our manuscript. The responses to the comments are as follows. For your kind information, we have carefully dealt with the comments of the reviewers as follows: We hope the revised manuscript meets the standard of publication. Thank you!
Point 1: The revised version of the manuscript highlights much effort in the improvement.
The authors' responses are adequate from my point of view. Before publication, the authors should provide figures with a higher resolution (especially fig. 2, 5 and 6) and modify the reference list according to the journal guidelines..
Response: Thank you very much for the comments and suggestions,We have uploaded and provided higher resolution images (especially Figures 2, 5, and 6), and have revised the reference list according to the journal guide.
